# Explainable Pathomics Feature Visualization via Correlation-aware Conditional Feature Editing

**Yuechen Yang**[1] ⬤iD                 YUECHEN.YANG@VANDERBILT.EDU
**Junlin Guo**[1]                     JUNLIN.GUO@VANDERBILT.EDU
**Ruining Deng**[2]                    RUD4004@MED.CORNELL.EDU
**Junchao Zhu**[1]                   JUNCHAO.ZHU@VANDERBILT.EDU
**Zhengyi Lu**[1]                    ZHENGYI.LU@VANDERBILT.EDU
**Chongyu Qu**[1]                   CHONGYU.QU@VANDERBILT.EDU
**Yanfan Zhu**[1]                   YANFAN.ZHU@VANDERBILT.EDU
**Xingyi Guo**[3]                     XINGYI.GUO@VUMC.ORG
**Yu Wang**[3]                      YU.WANG.2@VUMC.ORG
**Shilin Zhao**[3]                    SHILIN.ZHAO.1@VUMC.ORG
**Haichun Yang**[3]                   HAICHUN.YANG@VUMC.ORG
**Yuankai Huo**[1]                   YUANKAI.HUO@VANDERBILT.EDU

[1] *Vanderbilt University, Nashville TN 37235, USA*

[2] *Weill Cornell Medicine, New York, NY 10065, USA*

[3] *Vanderbilt University Medical Center, Nashville TN 37232, USA*

**Editors:** Accepted for publication at MIDL 2026

## Abstract

Pathomics is a recent approach that offers rich quantitative features beyond what black-box deep learning can provide, supporting more reproducible and explainable biomarkers in digital pathology. However, many derived features (e.g., "second-order moment") remain difficult to interpret, especially across different clinical contexts, which limits their practical adoption. Conditional diffusion models show promise for explainability through feature editing, but they typically assume feature independence, an assumption violated by intrinsically correlated pathomics features. Consequently, editing one feature while fixing others can push the model off the biological manifold and produce unrealistic artifacts. To address this, we propose a Manifold-Aware Diffusion (MAD) framework for controllable and biologically plausible cell nuclei editing. Unlike existing approaches, our method regularizes feature trajectories within a disentangled latent space learned by a variational auto-encoder (VAE). This ensures that manipulating a target feature automatically adjusts correlated attributes to remain within the learned distribution of real cells. These optimized features then guide a conditional diffusion model to synthesize high-fidelity images. Experiments demonstrate that our approach is able to navigate the manifold of pathomics features when editing those features. The proposed method outperforms baseline methods in conditional feature editing while preserving structural coherence.

**Keywords:** Digital Pathology, diffusion models, pathomics, image generation, explainability.

## 1. Introduction

While deep learning models excel in predictive performance, pathomics remains valuable for its ability to decompose histological patterns into explicitly defined quantitative metrics such as shape, intensity, and texture. Tools like CellProfiler (McQuin et al., 2018; Stirling et al., 2021) and Pyspatial (Yang et al., 2025) allow researchers to extract hundreds of quantitative phenotypic features from pathology images. These high-dimensional signatures have supported clinical studies, for example in renal artery morphology (Yin et al., 2025b) and nephron-specific glomerular responses (Yin et al., 2025a). However, an interpretability gap remains. Statistical models can indicate which features are associated with an outcome, but clinicians still lack a clear picture of how a numerical change in a feature (for example, compactness increasing from 0.85 to 0.95) appears in the corresponding tissue object. Figure 1(a) sketches an ideal explanation system in which a user adjusts feature sliders and can immediately see how the object in pathology image changes.

**(a) Ideally:** An editing-based explanation AI could independently visualize features using conditional diffusion.

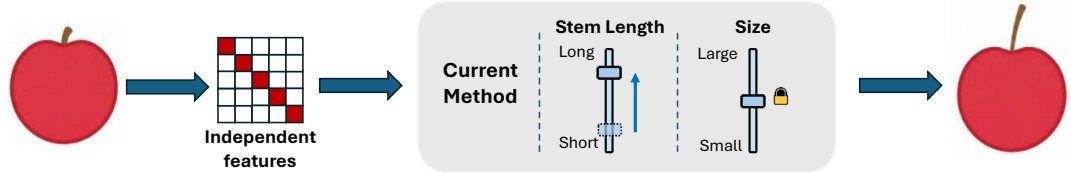

**(b) In reality:** Explaining pathomics features with such an AI is challenging because the features are often correlated, so editing a single feature may introduce conflicts.

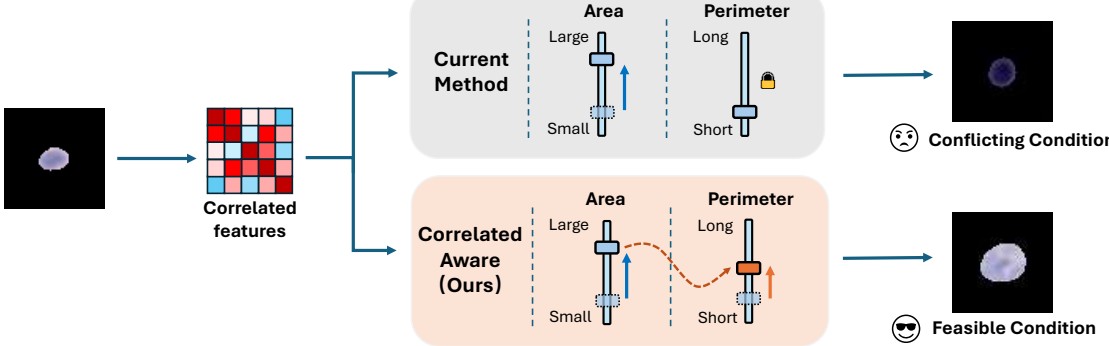

Figure 1: **Feature editing in pathomics.** (a) In generic object editing (e.g., an apple), attributes such as stem length and size operate independently. Modifying one feature does not impose constraints on others. (b) Pathomics features often exhibit intrinsic correlations (e.g., Area and Perimeter). The "Current Method" modifies the target feature (Area) while fixing the correlated feature (Perimeter), creating a geometric conflict and resulting in an infeasible edit. The proposed "Correlated-Aware" framework updates the correlated attribute alongside the target, ensuring the output remains within a feasible biological structure.

One way to approach such an explanation system is to use generative models that map features back to images. In this view, a generative model takes a feature vector as input, produces an image as output, and can act as a visual explanation tool. Recent work has explored this idea for pathomics. CP2Image (Ji et al., 2024) generates cell images directly from CellProfiler metrics, demonstrating that these hand-crafted features encode sufficient information for visual reconstruction. Continuous Conditional Diffusion Model (CCDM) (Ding et al., 2024) introduces mechanisms for conditioning diffusion models on continuous scalar values, enabling fine-grained control over attributes like cell counts or angles.

However, applying this idea to pathomics feature editing introduces a key difficulty. Many conditional pipelines implicitly rely on a feature independence assumption: when a target attribute such as area is modified, the remaining attributes are kept fixed. As noted in the CP2Image paper, current practices involve isolating a target feature for enhancement while keeping the remaining dimensions constant (Ji et al., 2024). Figure 1(b) illustrates why this is problematic for pathomics features. Attributes such as area and perimeter are intrinsically correlated and lie on a low-dimensional biological manifold. Increasing the area of a nucleus without adjusting its perimeter violates geometric laws and creates a conflicting condition for the generative model.

To address this, we propose Manifold-Aware Diffusion (MAD), a framework for controllable and correlation-aware pathomics feature editing. Instead of assuming that features can be edited independently, MAD performs feature editing on a learned manifold of pathomics features. The framework follows a two-stage design. First, a variational auto-encoder (VAE) (Kingma and Welling, 2013) learns the disentangled latent distribution of the features. Second, a conditional diffusion model synthesizes nucleus images conditioned on feature vectors. At inference time, when a user specifies a target feature value, MAD performs latent-guided optimization to obtain an edited feature vector that moves toward the target while adjusting correlated features so that the result stays on the learned manifold. We summarize our contributions as follows:

- **We propose MAD for explainable editing of correlated pathomics features.** MAD is a diffusion-based framework that edits continuous feature vectors and visualizes how changes in correlated features affect nucleus appearance.

- **MAD regularizes feature editing with a VAE-learned feature manifold.** The model performs edits within this manifold so that changes to a target feature are accompanied by correlated adjustments.

- **MAD scales to high-dimensional pathomics signatures.** A single model supports editing of 75-dimensional nuclear feature vectors and allows users to adjust each feature dimension individually.

## 2. Method

We propose MAD, a framework for editing correlated pathomics features with a conditional diffusion model. Figure 2 summarizes the overall idea of MAD. Edits are constrained

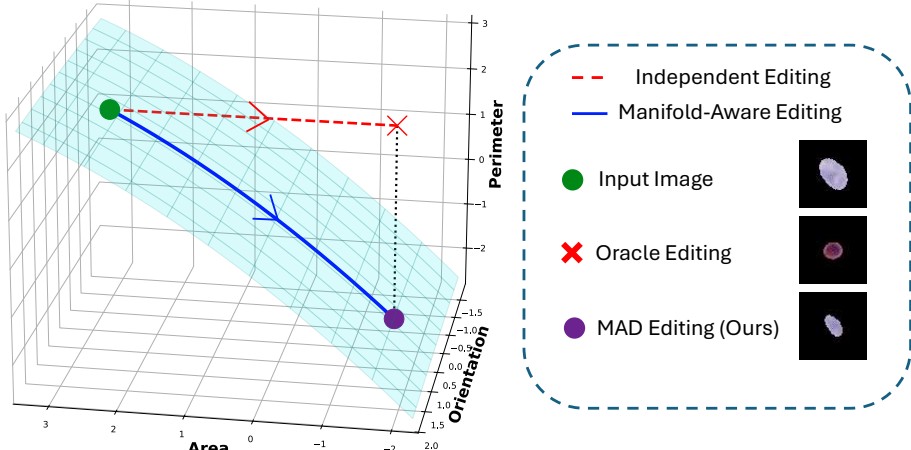

Figure 2: **Independent editing versus manifold-aware editing of correlated pathomics features**. The turquoise surface depicts the manifold of real nuclei in the feature space of area, perimeter, and orientation. The green point marks the input nucleus. To decrease area, independent editing (red dashed path) changes the area coordinate while keeping the other features fixed, reaching the oracle editing point (red cross), which lies off the manifold and leads to an out-of-distribution image. Manifold-aware editing (blue path) follows the manifold and jointly updates correlated features, reaching the MAD editing result (purple point) on the manifold. The images on the right show the input image, the image generated under the oracle editing condition, and the on-manifold result produced by our proposed model (MAD).

to follow a manifold of real nuclei features so that correlated attributes change together instead of moving off the manifold. To implement this idea, MAD decouples learning the image generator from learning the structure of the feature space, as shown in Figure 3. A conditional diffusion model learns to synthesize nucleus images from feature vectors, and a disentangled VAE learns a latent representation of valid feature combinations. At inference time, we perform latent-guided optimization in the VAE latent space to obtain an edited feature vector that moves toward a user-specified target while remaining on the learned manifold, and then use this feature vector as the condition for diffusion-based image editing.

## 2.1. Conditional Diffusion Model

We train a denoising diffusion probabilistic model (Ho et al., 2020) on single-nucleus images, which is illustrated in Figure 3(a). Each training sample consists of an image $x_0$ and its associated feature vector $y \in \mathbb{R}^N$ computed by a pathomics pipeline. This pairing ensures that the model learns to associate each image with the quantitative description that will be available in a downstream analysis pipeline. Following continuous conditional diffusion

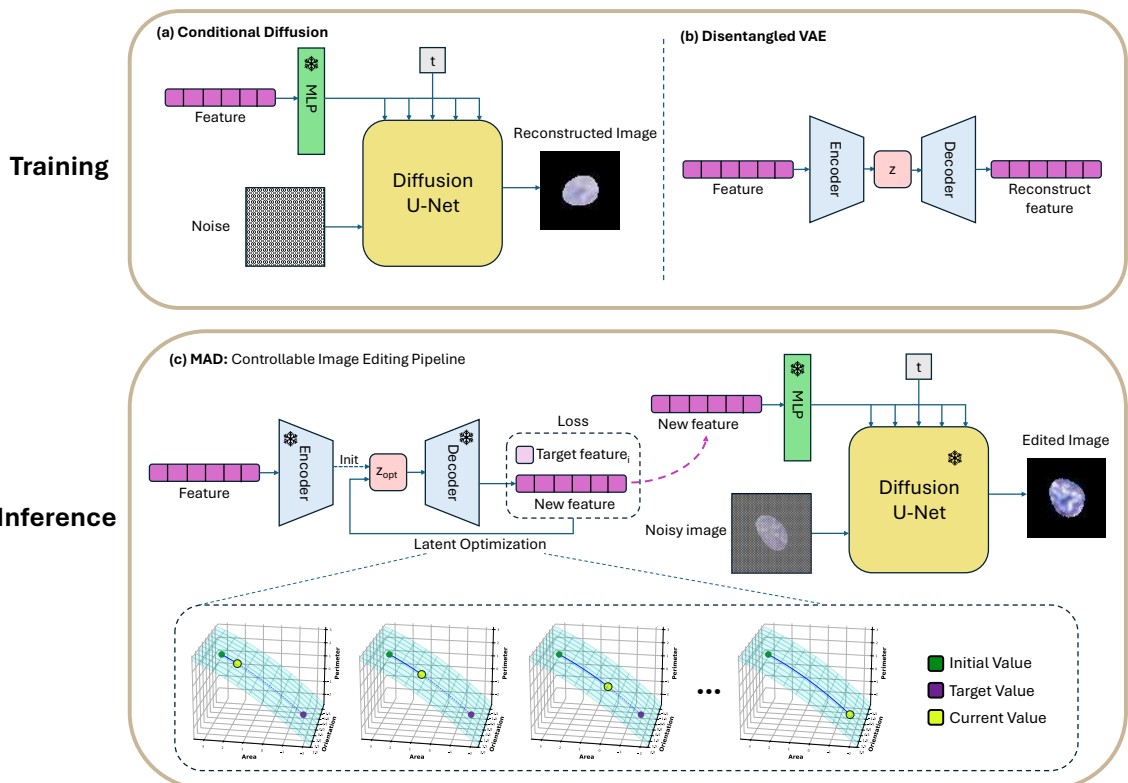

Figure 3: **Training and inference pipeline of MAD.** (a) A conditional diffusion model learns to use a feature vector to reconstruct the nucleus image. A pre-trained MLP encodes the feature vector into a conditioning embedding for the diffusion U-Net. (b) A disentangled VAE is trained on pathomics features. The encoder maps a feature vector to a latent variable $z$ and the decoder reconstructs the feature vector. (c) Given an input nucleus and a target feature value, the encoder initializes a latent variable from the feature of the input image. Through latent optimization, the decoded feature moves toward the target feature along the learned feature manifold. The bottom plots illustrate this trajectory in the feature space. The optimized feature vector is then used as the condition for the diffusion U-Net to edit the input image.

models, we encode $y$ with a multilayer perceptron (MLP) into a conditioning embedding $c(y)$ that is injected into each block of a U-Net denoiser (Ronneberger et al., 2015).

Let $q(x_t \mid x_0)$ denote the forward diffusion process that adds Gaussian noise to $x_0$ at time step $t$. The conditional U-Net $\epsilon_\theta$ is trained to predict the added noise given the noisy image, the time step, and the feature embedding:

$$\mathcal{L}_{\text{diff}} = \mathbb{E}_{x_0, y, \epsilon, t}\left[\|\epsilon - \epsilon_\theta(x_t, t, c(y))\|_2^2\right]. \tag{1}$$

At test time, we run the reverse diffusion process starting from Gaussian noise and repeatedly apply $\epsilon_\theta(\cdot, c(\tilde{y}))$ with a chosen conditioning feature vector $\tilde{y}$ to obtain a synthetic or edited nucleus image.

## 2.2. Feature Manifold VAE

Pathomics feature vectors contain correlated measurements such as area, perimeter, and shape descriptors. To model these correlations, we learn a latent representation of valid feature combinations with a $\beta$-VAE (Higgins et al., 2017), which is shown in Figure 3(b). The VAE encoder $\mathcal{E}$ maps a feature vector $y$ to a latent code $z \in \mathbb{R}^d$, and the decoder $\mathcal{D}$ reconstructs a feature vector $\hat{y}$ from $z$.

We train the VAE on feature vectors using the following objective:

$$\mathcal{L}_{\text{vae}} = \mathbb{E}_y \left[ \|y - \mathcal{D}(\mathcal{E}(y))\|_2^2 \right] + \beta \, D_{\text{KL}}\big(q(z \mid y) \,\|\, \mathcal{N}(0, I)\big), \qquad (2)$$

where $q(z \mid y)$ is the approximate posterior produced by the encoder and $\beta$ controls the strength of the Kullback–Leibler regularization. The latent space $\mathcal{Z}$ learned by the VAE provides a continuous feature manifold on which nearby latent codes correspond to feature vectors that are close in the original attribute space.

## 2.3. Latent-Guided Feature Editing

Given a trained diffusion model and VAE, MAD performs feature editing at test time through latent-guided optimization, as shown in Figure 3(c). Suppose an input nucleus image $x$ has measured feature vector $y_{\text{orig}}$ and we wish to change the $k$-th feature to a target value $v_{\text{tgt}}$. Our goal is to obtain an edited feature vector $y_{\text{new}}$ that reaches the target value on dimension $k$ while remaining consistent with $y_{\text{orig}}$ on other dimensions.

**Initialization on the feature manifold.** We first project $y_{\text{orig}}$ onto the VAE latent space using the encoder

$$z_{\text{init}} = \mathcal{E}(y_{\text{orig}}). \qquad (3)$$

This initialization places the optimization in a region of latent space that corresponds to observed nuclei features.

**Latent optimization.** We then optimize $z$ by gradient descent while keeping the VAE parameters fixed. The optimization objective balances matching the target feature and regularizing changes to the remaining features and the latent code:

$$\mathcal{L}_{\text{opt}}(z) = \lambda_{\text{tgt}}\big(\mathcal{D}(z)_k - v_{\text{tgt}}\big)^2 + \lambda_{\text{reg}} \sum_{j \neq k} w_j \big(\mathcal{D}(z)_j - y_{\text{orig},j}\big)^2 + \lambda_{\text{prior}}\|z\|_2^2. \qquad (4)$$

The first term encourages the decoded feature $\mathcal{D}(z)_k$ to match the target value on dimension $k$. The second term regularizes changes on non-target dimensions, where $w_j$ are per-feature weights that control the strength of this regularization. The third term keeps $z$ close to the origin of the Gaussian prior. During optimization we decode $\mathcal{D}(z)$ at each iteration; the bottom panel of Figure 3 illustrates the trajectory of the decoded features in a low-dimensional projection of the feature space.

**Diffusion-based image editing.** After optimization we obtain the edited feature vector

$$y_{\text{new}} = \mathcal{D}(z^*), \tag{5}$$

where $z^*$ is the final latent code. We feed $y_{\text{new}}$ into the same conditioning network $c(\cdot)$ used during diffusion training and run the reverse diffusion process to generate an edited image $\tilde{x}$. In all experiments, the diffusion parameters and the conditioning MLP are fixed during editing; only the latent code $z$ is updated at test time.

## 3. Data and Experiments

### 3.1. Data

**Dataset construction.** We construct a nucleus image dataset from 1,556 whole-slide images (WSIs) of kidney tissue at $40\times$ magnification. The WSIs include human and rodent samples stained with H&E, PAS, and PASM, drawn from public repositories (NEPTUNE (Barisoni et al., 2013), HuBMAP (Howard et al., 2020)) and an internal collection at Vanderbilt University Medical Center (VUMC). This combination of species and staining protocols exposes the model to a broad range of nuclear morphologies and staining appearances. From each WSI we randomly extract five $512 \times 512$ image patches. Nucleus instance segmentation is performed with an ensemble of three published models, Cellpose (Stringer et al., 2021), StarDist (Weigert and Schmidt, 2022), and CellViT (Hörst et al., 2024). We select image patches for which all three cell foundation models generate high-quality nuclei segmentation outcomes, based on rating criteria defined by a renal pathologist with 20 years of experience at VUMC. The details of the dataset construction and curation pipelines can be found in (Guo et al., 2025b,a). Then, for computational convenience, each nucleus in the selected $512 \times 512$ patch is cropped and resized into a $64 \times 64$ nucleus-centered image. Patches containing fragmented nuclei or obvious artifacts are removed, yielding an initial pool of 93,643 nucleus-centered $64 \times 64$ image patches.

### 3.2. Pathomics Feature Extraction

For each nucleus patch, we compute a pathomics feature vector using Pyspatial (Yang et al., 2025). The feature set contains area- and shape-related descriptors derived from the nucleus mask. Starting from the full set of output features, we remove non-phenotypic identifiers and location-dependent quantities, and drop features that are constant or have very low variance across the dataset. This yields a 75-dimensional feature vector for each nucleus.

To reduce the impact of extreme values, we treat the top and bottom 2.5% of observations along each feature dimension as outliers. Nuclei that fall outside this range on any feature are excluded. After this filtering, the dataset used for model training and evaluation contains 28,809 nuclei, each associated with a 75-dimensional feature vector. All models are trained on this dataset, and quantitative evaluation of editing performance is conducted on a randomly sampled subset of 300 nuclei.

### 3.3. Implementation Details

The VAE and the conditional diffusion model are trained independently, meaning there is no fixed order between the two training stages and they can even be trained in parallel.

For the VAE, we use the Adam optimizer with a batch size of 256 and train for up to 3,000 epochs with early stopping (patience of 50 epochs). The latent code $z$ has dimensionality $d = 16$, chosen based on a PCA analysis of the 75-dimensional feature space. For the conditional diffusion model, we use the Adam optimizer with a batch size of 128 and train for 20,000 iterations.

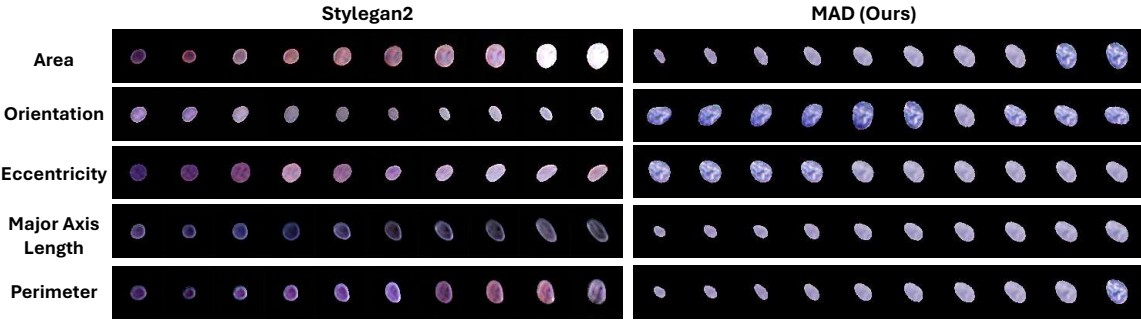

Figure 4: **Qualitative results between unconditional generation (StyleGAN2) and conditional generation (Ours).** Each row represents the traversal of a specific feature value from low to high. The left block shows StyleGAN2, which takes the target feature vector as input and generates a nucleus for each target value. The right block shows MAD, which edits a single input nucleus to match the same sequence of target values. StyleGAN2 samples follow the target feature but can introduce artifacts. MAD keeps the nuclei appearance close to the input image while changing the designated feature.

### 3.4. Metrics

We evaluate editing performance with three types of metrics. First, to verify that generated images represent valid single-nucleus structures, we compute the Segmentation Success Rate (SSR). Each generated image is binarized using a fixed intensity threshold of 10 (on a 0-255 scale), and connected component analysis is applied to the resulting mask. An image is considered successfully segmented if the mask contains one connected region. The SSR is defined as:

$$\text{SSR} = \frac{N_{\text{detected}}}{N_{\text{total}}} \times 100\%, \tag{6}$$

where $N_{\text{detected}}$ is the number of images with a single connected region and $N_{\text{total}}$ is the total number of generated images.

Second, to assess control accuracy, we measure agreement between the target feature value and the feature extracted from the edited image using Mean Absolute Error (MAE) and coefficient of determination ($R^2$). For each successfully segmented image, we extract the nucleus mask and compute pathomics features using Pyspatial (Yang et al., 2025), the same pipeline used to extract features from the training data. This ensures that evaluation is performed using actual pathomics measurements rather than learned proxies. Third, for iden-

tity preservation, we compute Learned Perceptual Image Patch Similarity (LPIPS) (Zhang et al., 2018) between the edited image and the input image. LPIPS computes distances in a deep feature space and is less sensitive than pixel-wise metrics such as SSIM to small spatial misalignments caused by shape changes.

### 3.5. Baselines

We compare MAD against two categories of baselines. As a generative model baseline, we use StyleGAN2-ADA (Karras et al., 2020) fine-tuned on nuclei images. At test time, we perform latent optimization in the StyleGAN $w$ space guided by a frozen ResNet-based feature regressor so that the predicted feature value of the generated image moves towards the target value. As editing model baselines, we use Stable Diffusion v1.5 (Rombach et al., 2022) with LoRA (Hu et al., 2022) fine-tuning and a variant of our method without the VAE module (MAD w/o VAE), which uses the target feature vector directly as the diffusion condition. All models are trained and evaluated on the same dataset split and feature configuration.

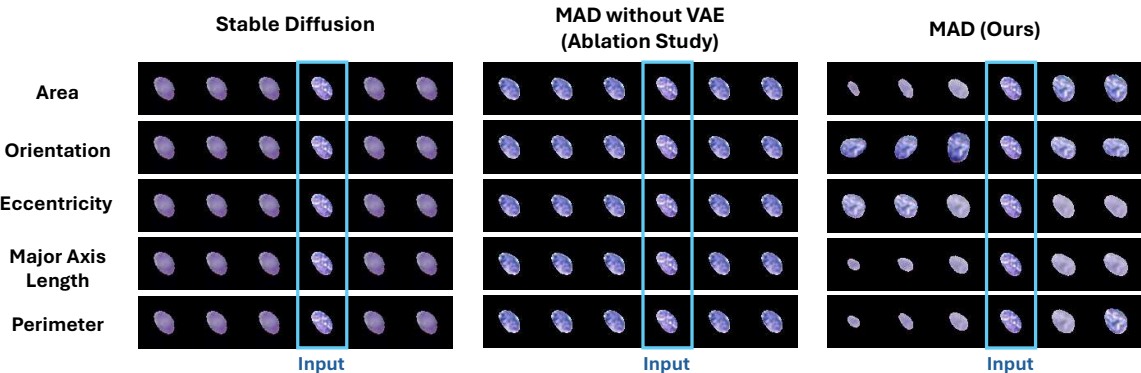

Figure 5: **Qualitative results for conditional editing.** Each row specifies a target trajectory for one nucleus feature. All three models take the same input nucleus (blue box) and the same sequence of target feature values. For Stable Diffusion and MAD without VAE, the edited nuclei stay close to the input nucleus across target values, which indicates that the conflicting conditions are not effectively reflected in the images. In contrast, MAD produces edited nuclei that follow the target feature change while preserving the overall appearance of the input nucleus.

### 3.6. Geometric Simulation

To quantitatively evaluate whether different methods can preserve known feature correlations during editing, we design a geometric simulation dataset with controlled mathematical relationships. We generate 4,000 synthetic images of size $64 \times 64$, each containing a single ellipse with a fixed aspect ratio of $a/b = 2$. This constraint establishes a deterministic

relationship between area and perimeter. Each ellipse is assigned a random intensity value that is independent of its geometric properties.

This design allows us to evaluate three aspects of feature editing: (1) editing accuracy, whether the target feature reaches the specified value; (2) correlation preservation, whether correlated features (area and perimeter) change together according to their mathematical relationship; and (3) independence preservation, whether independent features (intensity) remain unchanged when editing geometric properties. We use a delta-delta correlation plot to visualize the relationship between expected feature changes ($\Delta_{\mathrm{GT}} = y_{\mathrm{target}} - y_{\mathrm{origin}}$) and actual measured changes ($\Delta_{\mathrm{Pred}} = y_{\mathrm{final}} - y_{\mathrm{origin}}$). For correlated features, we report $R^2$ to measure how well actual changes follow expected changes. For independent features, we report MAE to measure deviations from the expected zero change.

## 4. Results

### 4.1. Qualitative Results

We first compare MAD with a StyleGAN2-based baseline that does not use an explicit feature manifold. Figure 4 shows StyleGAN2 and MAD along one-dimensional trajectories for five nucleus features: area, orientation, eccentricity, major axis length, and perimeter. For StyleGAN2, we generate one synthesized nucleus for each target feature value using latent optimization guided by a feature regressor. For MAD, we fix an input nucleus image and edit its feature vector along the same sequence of target values. In all rows, StyleGAN2 samples follow the direction of change in the target feature but also exhibit variation in other aspects of appearance, such as texture and intensity, across the row. In contrast, MAD edits modify the designated feature while keeping the overall staining pattern and chromatin texture tied to the original nucleus.

We next compare MAD with diffusion-based editing baselines that use feature conditions without an explicit manifold model. Figure 5 shows Stable Diffusion, MAD without the VAE component, and the full MAD model under the same feature trajectories as in Figure 4. All three methods take an input nucleus image and target feature values. For Stable Diffusion and MAD without VAE, the edited images along each row remain visually close to the highlighted input image, and the nucleus morphology changes only slightly as the target value varies. In contrast, MAD produces changes along each feature trajectory in Figure 5. When editing area or major axis length, the nucleus size increases or decreases along the row while the overall appearance of the nucleus remains approximately similar to the input. When editing orientation or eccentricity, the nucleus rotates or changes elongation while nearly maintaining texture and staining. These qualitative observations indicate that MAD responds to the target feature condition in correlated-feature settings.

### 4.2. Quantitative Results

We evaluate editing using four quantitative metrics that capture complementary aspects of performance. SSR measures whether generated images contain valid single-nucleus structures. MAE and $R^2$ quantify how closely the measured feature value of the edited image matches the prescribed target feature value. LPIPS measures perceptual difference between the edited image and the input image. All of these metrics are computed only on success-

Table 1: **Quantitative comparison of segmentation quality, feature control, and perceptual similarity.** The table reports SSR, MAE, $R^2$, and LPIPS for Style-GAN2 (unconditional generation) and for Stable Diffusion, MAD (w/o VAE), and MAD (conditional editing). For SSR and $R^2$, larger values indicate better performance. For MAE and LPIPS, smaller values indicate better performance. MAE, $R^2$, and LPIPS are computed only on successfully segmented images.

| Type | Method | SSR(%) ↑ | MAE ↓ | $R^2$ ↑ | LPIPS ↓ |
|---|---|---|---|---|---|
| Unconditional Generation | StyleGAN2 | 75.12 | 0.411 | 0.777 | 0.145 |
| Conditional Editing | Stable Diffusion | **100** | 1.756 | −0.660 | **0.036** |
| | MAD (w/o VAE) | **100** | 1.656 | −0.419 | 0.048 |
| | **MAD (Ours)** | 99.97 | **0.255** | **0.938** | 0.080 |

\* Note that StyleGAN2 does not support conditional editing.

fully segmented images using pathomics features extracted by Pyspatial. Together, these metrics summarize segmentation quality, feature control accuracy, and visual fidelity. The results are reported in Table 1.

StyleGAN2 achieves an SSR of 75.12%, indicating that approximately one quarter of generated images contain artifacts that prevent valid single-nucleus segmentation. In contrast, all conditional editing methods achieve SSR above 99%, with Stable Diffusion and MAD (w/o VAE) at 100% and MAD at 99.97%. For Stable Diffusion and MAD (w/o VAE), MAE is large (1.756 and 1.656) and $R^2$ is negative (−0.660 and −0.419), while LPIPS remains small (0.036 and 0.048). MAD achieves the lowest MAE (0.255) and highest $R^2$ (0.938) among all methods while maintaining an LPIPS of 0.080.

### 4.3. Geometric Simulation Results

We evaluate all methods on the geometric simulation dataset to assess editing accuracy, correlation preservation, and independence preservation. Detailed delta-delta correlation plots are provided in Appendix B.

When editing geometric features (Area or Perimeter), MAD achieves high editing accuracy ($R^2 > 0.99$) while simultaneously preserving the mathematical relationship between Area and Perimeter ($R^2 > 0.98$) and maintaining the independent Intensity feature (MAE $< 0.21$). StyleGAN2 shows comparable editing accuracy for geometric features ($R^2 > 0.97$) but exhibits large deviations in Intensity (MAE $> 1.4$). Stable Diffusion and MAD (w/o VAE) show lower editing accuracy for geometric features ($R^2 < 0.88$).

When editing Intensity, MAD achieves $R^2 = 0.998$ while preserving both geometric features (MAE $< 0.07$). StyleGAN2 achieves $R^2 = 1.000$ for Intensity but exhibits large deviations in geometric features (MAE $> 1.3$).

## 5. Discussion

We select baselines to cover different generative paradigms and isolate the contribution of each component. StyleGAN2-ADA represents GAN-based unconditional generation with post-hoc feature control via latent optimization. Stable Diffusion with LoRA represents diffusion-based conditional editing. MAD without VAE serves as an ablation to directly evaluate the contribution of the VAE-learned feature manifold.

The experimental results reveal distinct behaviors between editing models and unconditional generative models under conflicting conditions. Stable Diffusion and MAD (w/o VAE), as editing models designed to preserve input appearance, show limited editing capability when conditioned on geometrically infeasible feature combinations, as reflected by their negative $R^2$ values despite low LPIPS in Table 1. In contrast, StyleGAN2 as an unconditional generative model exhibits greater flexibility. However, this flexibility comes at the cost of image quality: StyleGAN2 produces artifacts when conditioned on infeasible feature combinations, resulting in a lower SSR of 75.12% compared to above 99% for all editing methods (Table 1). MAD resolves this trade-off by projecting edits onto a learned feature manifold, achieving both high editing accuracy and high image quality by ensuring that conditioning signals remain within the distribution of valid feature combinations. One limitation of the current framework is its inference speed: with 500 optimization steps, the latent-guided optimization requires approximately 17 seconds per edit, which may limit real-time interactive applications. Future work could explore amortized inference strategies to accelerate this process, as well as extending the framework to different nucleus types such as neoplastic, inflammatory, and epithelial cells.

## 6. Conclusion

In this paper, we present MAD, a manifold-aware diffusion framework for editing correlated pathomics feature vectors. MAD combines a VAE-learned feature manifold with a conditional diffusion model and applies latent-guided optimization at test time to adjust target features while allowing correlated features to change jointly.

Experiments on nucleus images indicate that MAD achieves feature control comparable to an unconditional generative baseline. At the same time, it preserves more of the input nucleus appearance than this baseline and responds more strongly to target feature changes than diffusion-based editing baselines. These results suggest that manifold-aware editing can support visual explanations of quantitative pathomics features and can help build interactive tools that link numerical feature changes to image-level morphology.

## Acknowledgments

This research was supported by NIH R01DK135597 (Huo), DoD HT9425-23-1-0003 (HCY), NSF 2434229 (Huo) and KPMP Glue Grant. This work was also supported by Vanderbilt Seed Success Grant and Vanderbilt-Liverpool Seed Grant. This research was also supported by NIH grants R01EB033385, R01DK132338. We extend gratitude to NVIDIA for their support by means of the NVIDIA hardware grant.

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

## Appendix A. Analysis of VAE Latent Space Structure

To verify that the VAE learns a meaningful manifold structure suitable for feature editing, we conduct three analyses on the learned latent space: dimensionality analysis, distance preservation, and interpolation smoothness.

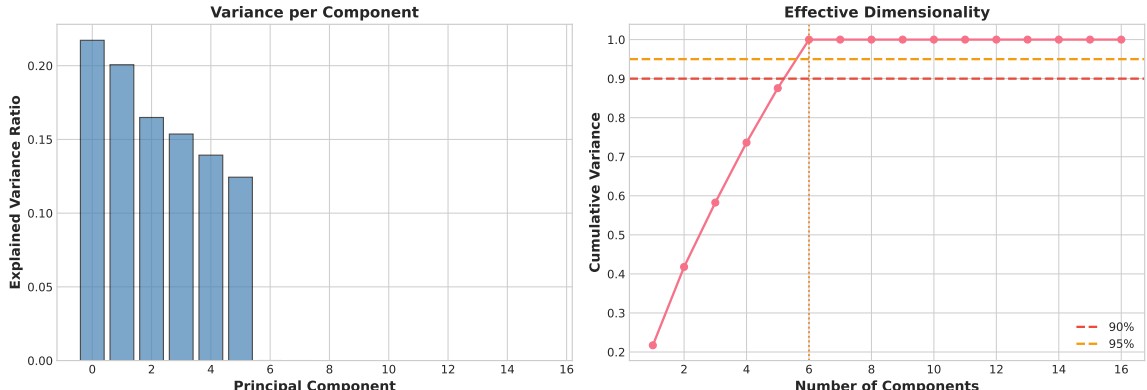

Figure 6: **Dimensionality analysis of the VAE latent space.** Left: Explained variance ratio for each principal component of the 16-dimensional latent codes. Right: Cumulative explained variance as a function of the number of components. Six dimensions are approximately to capture 90% of the variance, indicating that the VAE learns a compact, low-dimensional structure.

We apply PCA to the latent codes $z$ obtained by encoding the training feature vectors. Figure 6 shows that six principal components capture approximately 90% of the cumulative variance in the 16-dimensional latent space. This indicates that the VAE compresses the 75-dimensional feature space into a structured representation with low effective dimensionality, consistent with the hypothesis that pathomics features lie on a low-dimensional manifold.

A well-structured latent space should preserve relative distances: nearby points in latent space should correspond to similar feature vectors. We randomly sample pairs of feature vectors from the dataset, encode them into latent codes, and compare pairwise distances in latent space with pairwise distances in the decoded feature space. Figure 7 shows a strong linear relationship between these distances, with a Pearson correlation of 0.844. This confirms that the VAE learns a smooth mapping that preserves the neighborhood structure of the original feature space.

For manifold-aware editing, it is essential that linear interpolation in latent space produces smooth, continuous changes in the decoded features. We randomly select pairs of latent codes and linearly interpolate between them with 50 steps. For each interpolated latent code, we decode it to obtain a feature vector and track how individual features change along the interpolation path. Figure 8 shows that all features change smoothly along these paths, with an average smoothness score (mean squared second derivative) of 0.0026. This smooth behavior supports the use of latent space optimization for controlled feature editing.

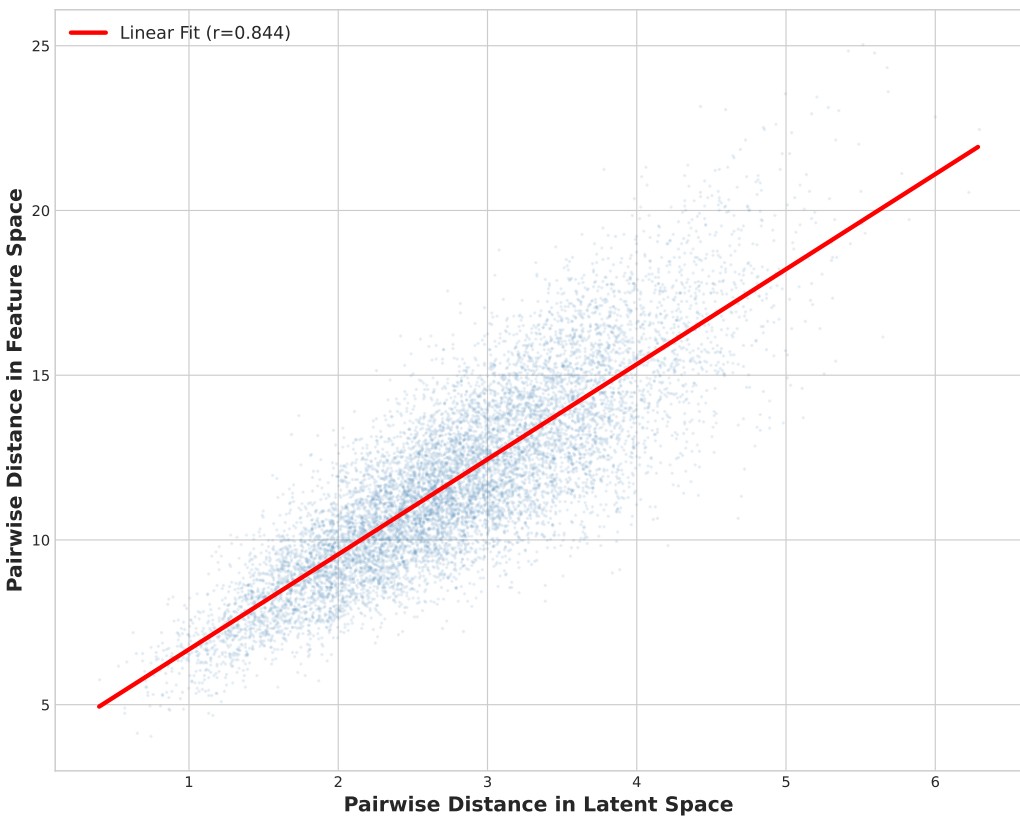

Figure 7: **Distance preservation between latent space and feature space.** Each point represents a pair of nuclei. The x-axis shows the Euclidean distance between their latent codes, and the y-axis shows the Euclidean distance between their decoded feature vectors. The Pearson correlation coefficient is 0.844, indicating that the VAE preserves relative distances during encoding and decoding.

Together, these analyses provide evidence that the VAE latent space captures a meaningful, low-dimensional manifold structure that preserves distances and supports smooth interpolation, justifying its use for manifold-aware feature editing.

## Appendix B. Geometric Simulation Details

Figure 9 shows the complete delta-delta correlation plots for the geometric simulation experiment. Each subplot compares expected feature changes (x-axis) with actual measured changes (y-axis). Rows correspond to the edited feature (Area, Perimeter, Intensity), and columns correspond to the measured feature. Diagonal entries evaluate editing accuracy; off-diagonal entries evaluate correlation preservation (for Area-Perimeter) or independence preservation (for Intensity). Points falling on the $y = x$ diagonal line indicate perfect edit-

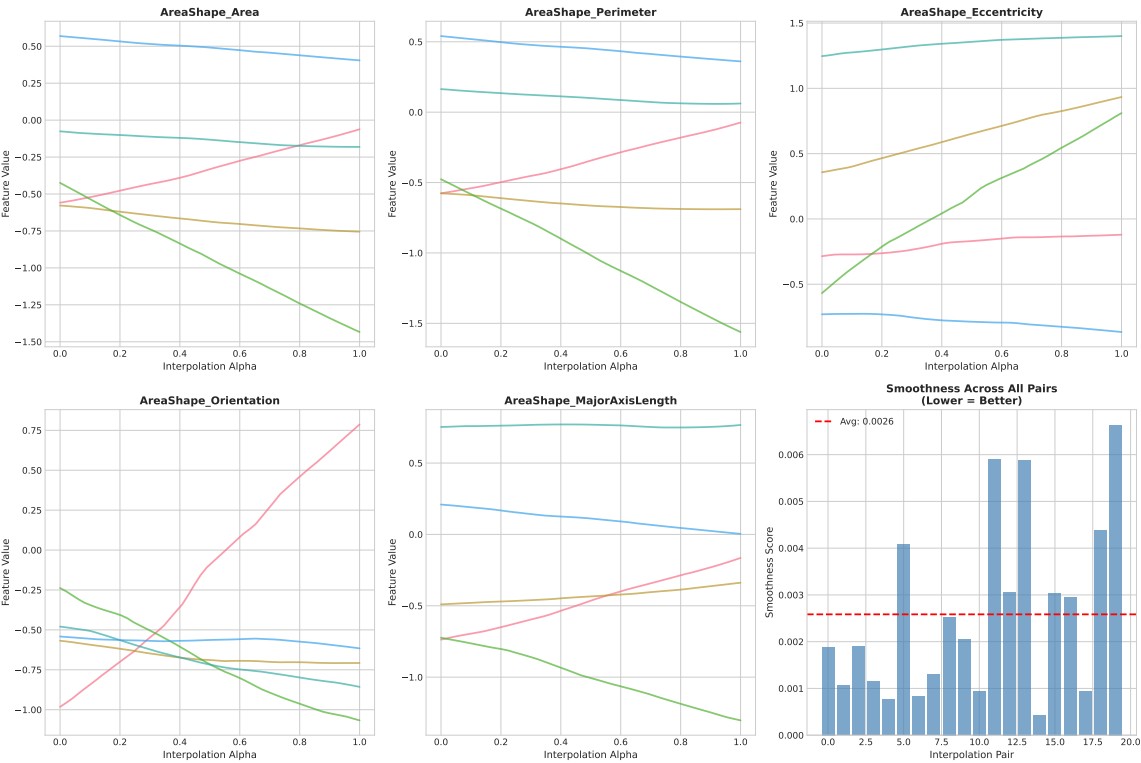

Figure 8: **Latent space interpolation produces smooth feature trajectories.** Each
subplot shows how a specific pathomics feature changes as we linearly interpo-
late between two latent codes. Different colors represent different interpolation
pairs. The bottom-right panel shows the smoothness score (mean squared sec-
ond derivative) for each pair, with an average of 0.0026, indicating that feature
trajectories are smooth rather than discontinuous.

ing. $R^2$ is reported for editing accuracy and correlation preservation; MAE is reported for
independence preservation.

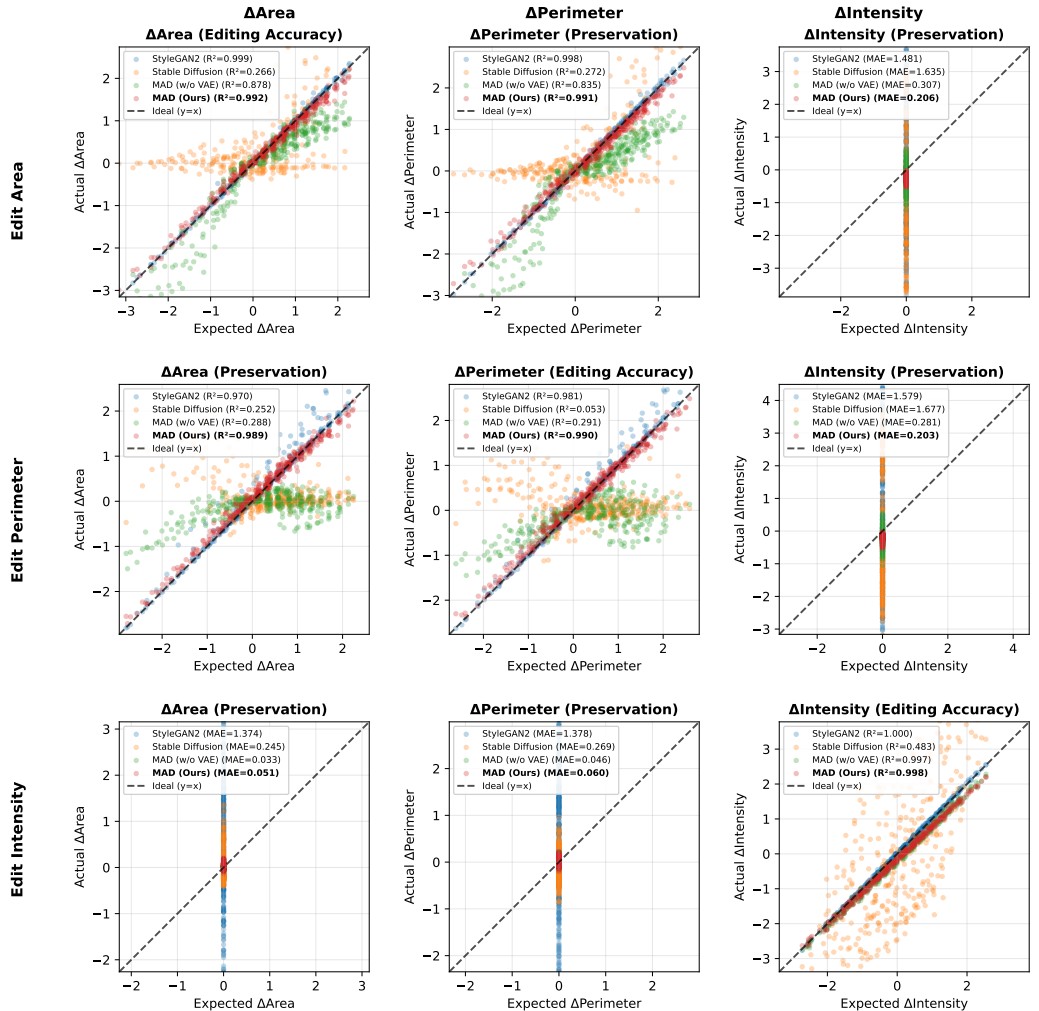

Figure 9: **Geometric simulation results.** Delta-delta correlation plots showing expected versus actual feature changes across all methods. When editing Area (top row), MAD achieves $R^2 = 0.992$ for the target feature, $R^2 = 0.991$ for the correlated Perimeter, and MAE = 0.206 for the independent Intensity. When editing Perimeter (middle row), MAD achieves $R^2 = 0.990$, $R^2 = 0.989$, and MAE = 0.203 respectively. When editing Intensity (bottom row), MAD achieves $R^2 = 0.998$ while preserving Area (MAE = 0.051) and Perimeter (MAE = 0.060). StyleGAN2 shows high editing accuracy but larger deviations in independent features. Stable Diffusion and MAD (w/o VAE) show lower editing accuracy for geometric features.

