# OpenReview forum: "Explainable Pathomics Feature Visualization via Correlation-aware Conditional Feature Editing"
_MIDL.io/2026/Conference — MIDL 2026 Poster_

### Official Review · Reviewer_1zAK · 2026-01-07

**Confidence:** 3
**Preliminary Rating:** 4

**Summary:**

This work addresses the anatomically-relevant editing of pathomics features by introducing a Manifold-Aware Diffusion (MAD) framework for biologically plausible cell nuclei editing. The key idea is to avoid unrealistic edits caused by correlated features by learning a disentangled latent manifold with a VAE. Edited features are constrained to remain on the biological manifold and are then used to guide a conditional diffusion model for image synthesis. Experiments show that MAD navigates the pathomics feature manifold more effectively than existing methods, producing high-fidelity images while preserving structural coherence.

**Strengths:**

The article is well-written and well-structured. It investigates the interesting problem which is of the high value to the community, and, moreover, can be potentially implemented for other medical fields. Authors show the novel solution and carefully compared it with baselines.

**Weaknesses:**

It's not clear why 'The proposed method outperforms baseline methods in conditional feature editing while preserving structural coherence.'.
Table 1 shows that StyleGAN is better in terms of MAE and R2, and SD is better in terms of LPIPS.

**Detailed Comments:**

'is trained to predict the added noise given the noisy image' - Figure 3 shows the noisy image during inference, not training
It would be great to make the code publicly available.

**Justification Of The Preliminary Rating:**

The authors showed the novel pipeline for the very practical problem, clearly described their motivation and pipeline's steps. Finally, they visualised the results of the method and compared its performance with baselines.

**Questions To Address In The Rebuttal:**

Why is the result of VAE the manifold? how can it be proven?
Information about training - what were optimizers, number of epochs? What was the order of training two parts?

---

> ### Author Response · Authors · 2026-01-25
> **Responses to Reviewer 1zAK**
>
> We sincerely appreciate your review and the constructive comments you provided. We have addressed your concerns with a point-by-point response.
>
> ---
> > **R4Q1:** It's not clear why 'The proposed method outperforms baseline methods in conditional feature editing while preserving structural coherence.'. Table 1 shows that StyleGAN is better in terms of MAE and R2, and SD is better in terms of LPIPS.
>
> Thank you for this important question. The revised *Table 1* (with updated evaluation using actual pathomics re-measurement) shows that MAD achieves the best MAE (0.255) and $R^2$ (0.938) among all methods.
>
> The key insight is that methods must be compared within their respective categories:
>
> 1. StyleGAN2 (unconditional generation): Achieves reasonable feature control but only 75.12% SSR—approximately 25% of generated images contain artifacts that prevent valid segmentation. StyleGAN2 cannot preserve input appearance since it has no input image.
>
> 2. Stable Diffusion and MAD w/o VAE (conditional editing): Achieve 100% SSR and low LPIPS (good appearance preservation), but negative $R^2$ values indicate they fail to achieve target features when conditions are geometrically infeasible.
>
> 3. MAD (Ours): Achieves 99.97% SSR, the best feature accuracy ($R^2 = 0.938$), and reasonable LPIPS (0.080). MAD resolves the trade-off by projecting edits onto the learned feature manifold.
>
> As stated in the revised *Discussion*: "MAD resolves this trade-off by projecting edits onto a learned feature manifold, achieving both high editing accuracy and high image quality by ensuring that conditioning signals remain within the distribution of valid feature combinations."
>
>
>
> ---
> > **R4Q2:** 'Is trained to predict the added noise given the noisy image' - Figure 3 shows the noisy image during inference, not training
>
>
> Thank you for noting this potential confusion. The statement in *Section 2.1* describes the training objective of the diffusion model, where the U-Net learns to predict noise given a noisy image. *Figure 3(c)* shows the inference pipeline, where the trained model is applied to edit images. The noisy image in the inference diagram is the starting point of the reverse diffusion process (initialized from the input image with added noise), which is then iteratively denoised using the trained model. This is standard diffusion model practice where training and inference both involve noisy images but in different roles.
>
> ---
> > **R4Q3:** It would be great to make the code publicly available.
>
> We appreciate this suggestion and are committed to open science. We plan to release the code publicly upon completion of the current research phase, as we are actively using the codebase for ongoing projects. We will publish the repository upon the conclusion of this phase.
>
>
> ---
> > **R4Q4:** Why is the result of VAE the manifold? how can it be proven?
>
>
> We thank the reviewer for requesting validation of the manifold hypothesis. We have added a VAE latent space analysis in the new *Appendix A*. Three analyses support the manifold structure:
> 1. Dimensionality analysis: PCA shows 6 principal components capture 90% of variance in the 16-dimensional latent space, indicating compact structure.
>
> 2. Distance preservation: Pearson correlation of 0.844 between latent space distances and feature space distances confirms the VAE preserves neighborhood structure.
>
> 3. Interpolation smoothness: Linear interpolation in latent space produces smooth feature trajectories (average smoothness score = 0.0026), supporting the use of gradient-based optimization for controlled editing.
>
>
> ---
> > **R4Q5:** Information about training - what were optimizers, number of epochs? What was the order of training two parts?
>
> We appreciate the reviewer's suggestions for reproducibility details. We have added this information in the new *Section 3.3* (Implementation Details):
>
> - VAE: Adam optimizer, batch size 256, up to 3,000 epochs with early stopping (patience 50), latent dimension $d = 16$
> - Diffusion model: Adam optimizer, batch size 128, 20,000 iterations
> - Training order: "The VAE and the conditional diffusion model are trained independently, meaning there is no fixed order between the two training stages and they can even be trained in parallel."
>
> ---

---

### Official Review · Reviewer_V1Mc · 2026-01-10

**Confidence:** 3
**Preliminary Rating:** 3
**Final Rating:** 4

**Summary:**

The study introduces Manifold-Aware Diffusion (MAD) framework that enables control over the editing process as well as the correlation of various pathomics features, allowing for edits of feature sets with an understanding of how those features are related to each other and their effect on nuclear morphology through visualisation. Unlike previous work that merely addressed independent editing of features, MAD edits features within a learned manifold of pathomics representations.

**Strengths:**

1. The visualization of features is an important area of challenge and research within H&E image based prognosis. In this context, the proposed approach develops a common solution for addressing interpretability problems associated with the models developed.
2. Using a conditional diffusion-based paradigm to manipulate all interactions between different feature vectors, this approach allows for the editing of continuous feature vectors to show how changes to both individual and multiple features will cause some changes in how nuclei appear together.

**Weaknesses:**

1. One important limitation is that the only software package used for feature extraction (PySpatial) provides area and shape features as a basis for feature extraction. Because there are many more meaningful and informative features based on the biology and probability of the tumors (like tumor-infiltrating lymphocytes (TIL) maps), it is not possible to say whether the framework's approach would be effective for extending the framework beyond area and shape features.
2. In addition, the authors do not compare their proposed framework to typical methods for controlled image generation using conditional genAI models such as ControlNet and Multi-ControlNet. This is an important omission because these types of generative models can provide a useful context for comparisons to other generative approaches.
3. The goal of the framework is visualization, but the authors do not provide any evaluation by pathologists of the accuracy, appropriateness, or clinical significance of the images being generated. A combination of qualitative feedback from experts and scoring would greatly improve the authors' ability to substantiate their claims.
4. Furthermore, the paper does not adequately define how the framework can generalize across all types of nuclei, such as neoplastic, inflammatory, epithelial, dead, and connective/soft tissue cells. Providing some form of qualitative analysis for each type of nucleus would improve the overall crux of the evaluation.

**Detailed Comments:**

The authors claim that they chose image patches on which all three models provided high quality segmentation results according to criteria defined by a renal pathologist at VUMC. How were the decisions made? Was there any scoring method? Detailed explanation of this curating approach should be made more prominent within the experimental design descriptions.

**Justification Of Final Rating:**

The authors have provided detailed clarifications to my questions, with the exception of R3Q3 regarding evaluation by pathologists. While the authors note that evaluation of generated image quality and clinical appropriateness would be valuable for clinical deployment and plan to pursue this in future work, I believe such qualitative evaluation should also be included in a methods paper at venues such as MIDL. Nevertheless, I recommend a score of `weak accept'.

**Justification Of The Preliminary Rating:**

The proposed work introduces an interesting and innovative approach to editing correlated pathomics features using diffusion processes and an interpretive learning mechanism, it has some major weaknesses that will likely prevent me from making a strong endorsement. Some of these weaknesses include very little diversity in the types of features used, lack of any comparisons with other conditional generative techniques, and lack of an expert clinical-validation component.

**Questions To Address In The Rebuttal:**

Please refer to the weakness section.

---

> ### Author Response · Authors · 2026-01-25
> **Responses to Reviewer V1Mc (1/2)**
>
> We sincerely appreciate your review and the constructive comments you provided. We have addressed your concerns with a point-by-point response.
>
> ---
> > **R3Q1:** One important limitation is that the only software package used for feature extraction (PySpatial) provides area and shape features as a basis for feature extraction. Because there are many more meaningful and informative features based on the biology and probability of the tumors (like tumor-infiltrating lymphocytes (TIL) maps), it is not possible to say whether the framework's approach would be effective for extending the framework beyond area and shape features.
>
> We appreciate the reviewer's comment on feature selection. We focused on area and shape features because they exhibit strong intrinsic correlations (e.g., Area-Perimeter), which is precisely the challenge MAD is designed to address.
>
> To demonstrate generalization beyond area and shape features, our new geometric simulation experiment (*Section 3.6*) explicitly includes intensity as an independent feature alongside Area and Perimeter. Results show MAD correctly preserves intensity when editing geometric features (MAE $< 0.21$) and preserves geometry when editing intensity (MAE $< 0.07$). This suggests the framework can handle features with different correlation structures.
>
> The MAD framework is agnostic to the specific feature extraction pipeline—any continuous feature vector can be used as conditioning input. Extension to additional feature types (texture, intensity distributions) is a natural direction for future work.
>
>
> ---
> > **R3Q2:** The authors do not compare their proposed framework to typical methods for controlled image generation using conditional genAI models such as ControlNet and Multi-ControlNet.
>
> We appreciate the reviewer's suggestion. However, ControlNet addresses a different problem from ours:
>
> 1. Different conditioning types: ControlNet is designed for spatial map-conditioned generation (e.g., edge maps, depth maps), while our task is numeric feature-conditioned editing. Our input is a 75-dimensional pathomics feature vector (e.g., Area, Perimeter), not a spatial control map. There is no straightforward way to convert numeric features into the spatial conditions that ControlNet requires.
>
> 2. Does not address our core contribution: ControlNet does not address our core contribution—handling feature correlations during editing. When editing Area, correlated features like Perimeter should change accordingly. Our VAE manifold explicitly addresses this, while ControlNet has no mechanism for such correlation handling.
>
> 3. Our baselines validate the core claim: Our baselines (Stable Diffusion, StyleGAN2, and ablation without VAE) specifically evaluate whether manifold-aware editing outperforms naive direct editing, which is the central claim of our paper.
>
> Therefore, while ControlNet is an important technique, its design for spatial-map conditioning makes it less directly applicable to our numeric-feature editing task. We believe our current baselines more directly evaluate the core claim of this paper.
>
>
> ---
> > **R3Q3:** The goal of the framework is visualization, but the authors do not provide any evaluation by pathologists of the accuracy, appropriateness, or clinical significance of the images being generated. A combination of qualitative feedback from experts and scoring would greatly improve the authors' ability to substantiate their claims.
>
> We appreciate the reviewer emphasizing the importance of expert evaluation.  Our current evaluation focuses on quantitative validation of feature editing accuracy using actual pathomics re-measurement (*Section 3.4*). The new geometric simulation experiment provides additional validation with ground-truth feature relationships.
>
> Systematic pathologist evaluation of generated image quality and clinical appropriateness would be valuable for clinical deployment and is a direction we plan to pursue in future work.

---

> ### Author Response · Authors · 2026-01-25
> **Responses to Reviewer V1Mc (2/2)**
>
> > **R3Q4:** The paper does not adequately define how the framework can generalize across all types of nuclei, such as neoplastic, inflammatory, epithelial, dead, and connective/soft tissue cells. Providing some form of qualitative analysis for each type of nucleus would improve the overall crux of the evaluation.
>
> We thank the reviewer for this question regarding generalization. Our dataset is highly diverse, containing nuclei from human and rodent kidneys with multiple stains (H&E, PAS, PASM). While we did not explicitly classify nuclei into sub-types for this specific study, the MAD framework is class-agnostic. We have added a note in the *Discussion* that testing the framework on specific sub-types (e.g., neoplastic vs. inflammatory) is a planned direction for future work.
>
>
>
> ---
> > **R3Q5:** The authors claim that they chose image patches on which all three models provided high quality segmentation results according to criteria defined by a renal pathologist at VUMC. How were the decisions made? Was there any scoring method?
>
> Thank you for this question. Yes, a systematic scoring method was used. Foundation model predictions were evaluated using a rating-based system with three categories: "good" (capturing ~90% of nuclei), "medium" (50–90%), and "bad" (<50%), based on criteria defined by an expert renal pathologist. Two pathologist-trained students independently rated predictions in a blinded setting, with the renal pathologist reviewing cases where assessments differed. We selected image patches for which all three foundation models (Cellpose, StarDist, CellViT) received "good" ratings. The detailed methodology is described in [1].
>
>
> [1] Guo, Junlin, et al. "Evaluating cell AI foundation models in kidney pathology with human-in-the-loop enrichment." Communications Medicine 5.1 (2025): 495.

---

### Official Review · Reviewer_BLd3 · 2026-01-12

**Confidence:** 3
**Preliminary Rating:** 4

**Summary:**

This paper proposes MAD (Manifold-Aware Diffusion) for explainable pathomics feature visualization: given a nucleus image and a desired change in one quantitative pathomics feature (e.g., area), generate an edited nucleus that reflects that change without breaking correlated features (e.g., perimeter) that live on a biological manifold. The key idea is to learn a feature manifold with a β-VAE, then do latent-guided optimization in the VAE latent space so the edited feature vector stays plausible, and finally use a conditional diffusion model to synthesize the edited image.

**Strengths:**

The motivation is solid and specific: feature independence is a bad assumption for pathomics, and the paper explains the geometric failure mode clearly (area vs perimeter example is intuitive). The method is conceptually clean: separate “what feature combos are valid” (VAE manifold) from “how to render features into images” (conditional diffusion), then do test-time optimization in latent space. The experimental setup is fairly careful for this type of work: they compare against Stable Diffusion + LoRA, StyleGAN2-ADA + latent optimization, and an ablation MAD w/o VAE.

**Weaknesses:**

The evaluation depends on a pretrained ResNet regressor to measure edited features, which is a bit circular: the model is optimized to satisfy that regressor, so “control” might partly reflect regressor bias rather than true pathomics measured from masks. I’d like to see at least a small sanity check using actual feature extraction on re-segmented outputs where possible. The dataset pipeline is heavy (ensemble segmentation + expert rating + outlier filtering), which is fine, but it raises the question: does MAD stay stable when nuclei are messier (fragmented masks, staining artifacts), i.e., the exact cases clinicians care about?  StyleGAN2 is treated as “unconditional generation” so it’s not a true editing baseline; it’s more of a “best possible feature following” reference.

**Detailed Comments:**

Please clarify whether the edited outputs are ever re-segmented and re-measured with the same pathomics pipeline (even on a small subset). If not, explain why it’s infeasible.
A short runtime note would help: test-time optimization over z could be slow; how many steps do you run and what’s the typical latency per edit?
The Stable Diffusion baseline seems to “under-edit” (very low LPIPS, poor R square). Is that a limitation of conditioning on continuous features, or specific to this LoRA setup? Some tuning discussion would reduce reviewer doubts.

**Justification Of The Preliminary Rating:**

Overall, I think this is a nice “explainability-through-editing” paper for pathomics where the correlated-feature issue is real and often glossed over. The VAE-manifold constraint is a reasonable way to prevent impossible edits, and the numbers suggest it actually fixes the main failure mode of diffusion editors here (they mostly keep the image unchanged). What holds me back from a stronger score is evaluation realism: feature control is measured through a regressor, and the dataset is curated enough that I’m not yet convinced the method behaves well under messy real-world artifacts.

**Questions To Address In The Rebuttal:**

Can you show pathomics re-measurement (not just regressor prediction) on edited outputs for a subset?
How sensitive is MAD to segmentation noise / imperfect masks (since features come from masks)?
Any evidence that the VAE latent space is meaningfully “disentangled” or at least structured enough to justify manifold editing in 75D?

---

> ### Author Response · Authors · 2026-01-25
> **Responses to Reviewer BLd3 (1/2)**
>
> We sincerely appreciate your review and the constructive comments you provided. We have addressed your concerns with a point-by-point response.
>
> ---
>
> > **R2Q1:** The evaluation depends on a pretrained ResNet regressor to measure edited features, which is a bit circular. Can you show pathomics re-measurement (not just regressor prediction) on edited outputs for a subset?
>
> Thank you for this important observation. We have revised our evaluation pipeline to address this concern. As stated in the revised *Section 3.4*: "For each successfully segmented image, we extract the nucleus mask and compute pathomics features using Pyspatial, the same pipeline used to extract features from the training data. This ensures that evaluation is performed using actual pathomics measurements rather than learned proxies."
> *Table 1* has been updated with results from this non-circular evaluation using actual pathomics re-measurement.
>
> ---
> > **R2Q2:** Does MAD stay stable when nuclei are messier (fragmented masks, staining artifacts), i.e., the exact cases clinicians care about?
>
> We appreciate this practical question regarding robustness. At this stage, our method is designed for morphological analysis. Our dataset construction follows the curation pipeline described in [1], where image patches are selected based on high-quality segmentation outcomes from an ensemble of three foundation models. This ensures the training and evaluation data contain well-segmented nuclei without fragmented masks or severe artifacts.
>
> We are now trying to apply this tool to real clinical scenarios and we completely agree that the robustness of MAD is a question we need to explore in the future.
>
> [1] Guo, Junlin, et al. "Evaluating cell AI foundation models in kidney pathology with human-in-the-loop enrichment." Communications Medicine 5.1 (2025): 495.
>
>
> ---
> > **R2Q3:** StyleGAN2 is treated as “unconditional generation” so it’s not a true editing baseline; it’s more of a “best possible feature following” reference.
>
> We agree that StyleGAN2 serves a different role than editing baselines. As clarified in the revised *Discussion*: StyleGAN2 represents "GAN-based unconditional generation with post-hoc feature control via latent optimization." It demonstrates that feature control is achievable but at the cost of image quality—StyleGAN2 achieves only 75.12% SSR (*Table 1*), indicating that ~25% of generated images contain artifacts when conditioned on conflicting features. In contrast, MAD achieves 99.97% SSR while maintaining high feature accuracy, demonstrating the advantage of manifold-aware editing.
>
> ---
> > **R2Q4:** A short runtime note would help: test-time optimization over z could be slow; how many steps do you run and what’s the typical latency per edit?
>
> We appreciate the reviewer's suggestion to include runtime information. We have added this information to the revised *Discussion* section: "with 500 optimization steps, the latent-guided optimization requires approximately 17 seconds per edit."
>
> ---
> > **R2Q5:** The Stable Diffusion baseline seems to “under-edit” (very low LPIPS, poor R square). Is that a limitation of conditioning on continuous features, or specific to this LoRA setup? Some tuning discussion would reduce reviewer doubts.
>
> We thank the reviewer for this insightful observation. The under-editing behavior of Stable Diffusion (and MAD w/o VAE) reflects their design as editing models that preserve input appearance. When conditioned on geometrically infeasible feature combinations (e.g., large Area with small Perimeter), these models face a conflict between feature accuracy and appearance preservation. They resolve this by prioritizing appearance preservation, resulting in low LPIPS but poor feature control (negative $R^2$).

---

> ### Author Response · Authors · 2026-01-25
> **Responses to Reviewer BLd3 (2/2)**
>
> > **R2Q6:** How sensitive is MAD to segmentation noise / imperfect masks (since features come from masks)?
>
> We appreciate this question. It is important to clarify that MAD does not take masks as input during inference. The model receives only pathomics feature vectors and the nucleus image. Masks are used solely during data preparation to extract features via Pyspatial.
> Regarding the dataset, our curation pipeline follows Guo et al. (2025), which selects patches where three foundation models (Cellpose, StarDist, CellViT) produce consensus high-quality segmentation, with quality criteria defined by an experienced renal pathologist with 20 years of experience. This filtering ensures that the training and evaluation data contain reliable nucleus boundaries.
>
>
> ---
> > **R2Q7:** Any evidence that the VAE latent space is meaningfully “disentangled” or at least structured enough to justify manifold editing in 75D?
>
> We thank the reviewer for requesting validation of the manifold hypothesis. We have added a VAE latent space analysis in the new *Appendix A*. Three analyses support the manifold structure:
> 1. Dimensionality analysis: PCA shows 6 principal components capture 90% of variance in the 16-dimensional latent space, indicating compact structure.
>
> 2. Distance preservation: Pearson correlation of 0.844 between latent space distances and feature space distances confirms the VAE preserves neighborhood structure.
>
> 3. Interpolation smoothness: Linear interpolation in latent space produces smooth feature trajectories (average smoothness score = 0.0026), supporting the use of gradient-based optimization for controlled editing.

---

### Official Review · Reviewer_Nk2s · 2026-01-16

**Confidence:** 4
**Preliminary Rating:** 3
**Final Rating:** 4

**Summary:**

The authors study how a conditional diffusion model can be used to visualize pathomics feature editing.

Standard pipelines implicitly rely on a feature independence assumption, but this can lead to conflicting / biologically impossible conditioning for pathomics features. To address this, they use a VAE-learned feature manifold to regularize the feature editing, thereby obtaining _"an edited feature vector that moves toward a user-specified target while remaining on the learned manifold"_.

The proposed approach, called MAD, is applied to a dataset of 28.8k nucleus-centered images of size 64x64, derived from a set of 1.5k WSIs.

MAD is compared to a purely generative baseline in the form of StyleGAN2, a Stable diffusion editing model, and a variant of MAD without the VAE module.

**Strengths:**

- The paper is well written overall and easy to follow, it contains basically not a single typo or similar issue.

- The proposed MAD approach is clearly described, it is conceptually quite simple and makes sense overall. Figure 3 gives a good overview.

- At least compared to the evaluated baselines, StyleGAN2 and a Stable diffusion editing model, the proposed MAD seems to strike a good balance in the trade-off between feature control and perceptual similarity.

**Weaknesses:**

- The experimental evaluation could be more extensive and convincing. In particular, it lacks any type of external validation.

- At least to me, it is quite unclear how the proposed approach should be applied in practice, what the concrete practical use-cases are, or how much utility it actually would provide in real-world applications.

**Detailed Comments:**

Questions/suggestions:
- Could you please expand on how this tool actually would be used in practice? What are the main, concrete practical use-cases? How would this be useful in real-world clinical applications and/or research?

- Compared to the evaluated baselines, it is clear that MAD strikes a better trade-off between feature control and perceptual similarity. But, are there no other, stronger baselines that would be more relevant to compare to? And, suppose that another model would achieve a slightly better MAE but a slightly worse LPIPS than MAD (or the other way around), which model would then be preferred in practice?

- _"All models are trained on this dataset, and quantitative evaluation of editing performance is conducted on a randomly sampled subset of 300 nuclei"_, does this mean that nuclei from the same WSI can be included in both the train and eval set? Would it not make more sense to hold out a separate set of WSIs, from which all nuclei images used for evaluation are extracted?

- Just out of curiosity, what is the dimension of the latent code z in the VAE?

- In Figure 5, you mark the input nucleus image. What is the input image for MAD in Figure 4?






Minor thing:
- Figure 1 caption: I think the quotation marks should be e.g. ``Current Method'' instead of what you currently use.

**Justification Of Final Rating:**

The other reviews are somewhat mixed, I don't think they give me any obvious reason to change my score at least.

The authors provided a solid rebuttal that addressed some of my concerns, my view of the paper is more positive overall now.

While it is still not quite clear to me how useful this actually would be in practice, and I therefore remain somewhat borderline on the paper, I think this is quite solid work overall that probably deserves to be accepted.

**Justification Of The Preliminary Rating:**

This is a well-written paper with a quite simple proposed method that makes sense overall, but the experimental evaluation could be more convincing, and it is not clear to me how useful this actually would be in practice. Therefore, I'm currently borderline on the paper, leaning slightly towards reject.

**Questions To Address In The Rebuttal:**

Please see "Weaknesses" and "Questions/suggestions" above.

---

> ### Author Response · Authors · 2026-01-25
> **Responses to Reviewer Nk2s (1/2)**
>
> We sincerely appreciate your review and the constructive comments you provided. We have addressed your concerns with a point-by-point response.
>
> ---
> > **R1Q1:** The experimental evaluation could be more extensive and convincing. In particular, it lacks any type of external validation.
>
> We appreciate the reviewer's suggestion to strengthen our evaluation. In the revised manuscript, we have added a new geometric simulation experiment (*Section 3.6* and *Section 4.3*) that serves as external validation with ground-truth feature correlations. Results (*Figure 9*, newly added) show MAD achieves $R^2 > 0.99$ for editing accuracy while preserving the mathematical correlation ($R^2 > 0.98$) and maintaining independent features (MAE $< 0.21$). This controlled experiment validates our method's ability to navigate feature correlations.
>
> ---
> > **R1Q2**: Could you please expand on how this tool actually would be used in practice? What are the main, concrete practical use-cases? How would this be useful in real-world clinical applications and/or research?
>
> We thank the reviewer for asking about the clinical utility of our tool. At this stage, we are now serving MAD as a tool for visual explanation of pathomics features. When a statistical model identifies a feature (e.g., "compactness") as clinically relevant, MAD allows users to visualize what a change in that feature looks like on actual nuclei. This bridges the gap between numerical biomarkers and visual interpretation.
> Also, we are working on applications that include: (1) simulating morphological trajectories between time points in longitudinal studies (e.g., predicting 2-week mouse kidney morphology from 1-week and 3-week observations), and (2) generating visual explanations for feature-based clinical predictions.
>
>
> ---
> > **R1Q3:** Are there no other, stronger baselines that would be more relevant to compare to?
>
>
> We appreciate the reviewer's query regarding baseline selection. To the best of our knowledge, there are no existing methods specifically designed for correlation-aware continuous feature editing in pathomics. We selected baselines to cover different generative paradigms. As stated in the revised *Discussion* section: "StyleGAN2-ADA represents GAN-based unconditional generation with post-hoc feature control via latent optimization. Stable Diffusion with LoRA represents diffusion-based conditional editing. MAD without VAE serves as an ablation to directly evaluate the contribution of the VAE-learned feature manifold."
>
> ---
> > **R1Q4:** Which model would be preferred if another achieves slightly better MAE but worse LPIPS?
>
> This is an excellent point regarding the balance between editing intensity and identity preservation. For visualization/explanation purposes, preserving input appearance (low LPIPS) while achieving reasonable feature control is preferable, as users need to see how a specific nucleus changes. For data augmentation, higher feature accuracy (low MAE) may be prioritized. MAD provides a balanced solution suitable for the primary use case of visual explanation.
>
> ---
> > **R1Q5:** "All models are trained on this dataset, and quantitative evaluation of editing performance is conducted on a randomly sampled subset of 300 nuclei", does this mean that nuclei from the same WSI can be included in both the train and eval set? Would it not make more sense to hold out a separate set of WSIs, from which all nuclei images used for evaluation are extracted?
>
> Thank you for this observation. Our dataset construction follows the pipeline in [1], where image patches are curated based on segmentation quality ratings from three foundation models. While patches from the same WSI may appear in both training and evaluation sets, our task differs fundamentally from classification or segmentation: during editing, the input feature vector is explicitly specified, and the model must generate an image matching that target. This design reduces the risk of memorization, as the model cannot simply reproduce training images.
>
> Furthermore, our new geometric simulation experiment uses entirely synthetic data with no overlap to the training set, providing an independent validation that confirms our findings.
>
> [1] Guo, Junlin, et al. "Evaluating cell AI foundation models in kidney pathology with human-in-the-loop enrichment." Communications Medicine 5.1 (2025): 495.

---

> ### Author Response · Authors · 2026-01-25
> **Responses to Reviewer Nk2s (2/2)**
>
> > **R1Q6:** What is the dimension of the latent code z in the VAE?
>
> We thank the reviewer for this detailed question. The latent dimension is $d = 16$, chosen based on PCA analysis of the 75-dimensional feature space. This is now specified in the new *Section 3.3 (Implementation Details)*.
>
> ---
> >**R1Q7:** In Figure 5, you mark the input nucleus image. What is the input image for MAD in Figure 4?
>
> We appreciate the reviewer's careful observation of our figures. The input image for MAD in Figure 4 is the same as in Figure 5 (the highlighted nucleus). We did not mark it in Figure 4 because StyleGAN2 is an unconditional generator—it has no input image, only input features. Figure 4 emphasizes that while StyleGAN2 can follow target features, it lacks consistent style across the trajectory since there is no reference image to preserve.
>
> ---
> > **R1Q8:** Figure 1 caption: I think the quotation marks should be e.g. ``Current Method'' instead of what you currently use.
>
> We thank the reviewer for catching this formatting error. We have corrected the quotation marks to the standard LaTeX format (``Current Method'') in the revised manuscript.

---

> > ### Comment · Reviewer_Nk2s · 2026-01-31
> >
> > Thank you for the response.
> >
> > I have read the other reviews and all author responses.
> >
> > The other reviews are somewhat mixed, I don't think they give me any obvious reason to change my score at least.
> >
> > The authors provided a solid rebuttal that addressed some of my concerns, my view of the paper is more positive overall now.
> >
> > While it is still not quite clear to me how useful this actually would be in practice, and I therefore remain somewhat borderline on the paper, I think this is quite solid work overall that probably deserves to be accepted.
> >
> > I have increased my score to "4: Weak accept".

---

### Author Rebuttal · Authors · 2026-01-25

**Rebuttal:**

We sincerely thank all the reviewers and Area Chairs for their constructive feedback, which has significantly helped improve the quality of the manuscript. We have modified the manuscript according to the reviews to address the common concerns and comments proposed by the reviewers (R1: Nk2s, R2: BLd3, R3: V1Mc, R4: 1zAK). **All modifications are highlighted in blue.**
The main modifications in the paper are:
1. **Pathomics Re-measurement Using Actual Feature Extraction**: We appreciate reviewer 2's concern about circular evaluation using a learned regressor. We have revised our evaluation pipeline. For each successfully segmented image, we now extract the nucleus mask and compute pathomics features using Pyspatial, the same pipeline used to extract features from the training data. This ensures that evaluation is performed using actual pathomics measurements rather than learned proxies. *Table 1* has been updated with the new evaluation results.
2. **New Geometric Simulation Experiment**: To provide additional external validation and demonstrate MAD's ability to handle feature correlations, we have added a new geometric simulation experiment (*Section 3.6* and *Section 4.3*).
We generate synthetic ellipse images following a deterministic mathematical relationship. Each ellipse also has a random intensity value independent of geometry. This controlled setting allows us to evaluate:
      - Editing accuracy: Whether the target feature reaches the specified value
      - Correlation preservation: Whether correlated features (Area-Perimeter) change together
      - Independence preservation: Whether independent features (Intensity) remain unchanged
3. **VAE Latent Space Analysis**: We appreciate reviewers' concern about the VAE latent space. We have added an analysis of the latent space structure in the new *Appendix A*. We conduct three analyses including Dimensionality, Distance preservation, and Interpolation smoothness.

For the point-to-point responses, please refer to "Responses to Reviewer XXXX" below each review.

**Supporting Material:**

/attachment/662fc1bf6d1ccb138f7b04b8eaf0aaa0ae8fc10d.pdf

---

### Meta-Review · Area_Chair_X8dT · 2026-02-07

**Recommendation:** Accept (Poster)
**Confidence:** 4

**Metareview:**

All reviewers agree to accept the paper. The author should revise the manuscript accordingly before camera-ready. Congrats!

---

### Decision · Program_Chairs · 2026-02-13

Accept (Poster)